# Alterations in innate immune responses of patients with chronic rhinosinusitis related to cystic fibrosis

Gustavo L. Rezende[1,2,3]⊙*, Marcio Nakanishi[2,4,5]⊙, Shirley C. P. Couto[2], Carmen L. F. S. Martins[2,6], André L. L. Sampaio[4], Lucas F. F. Albuquerque[3], Selma A. S. Kückelhaus[2,3], Maria I. Muniz-Junqueira[2,3]

**1** Hospital de Base, Brasília, Federal District, Brazil, **2** Laboratory of Cellular Immunology, Pathology, Faculty of Medicine, University of Brasilia, Brasília, Federal District, Brazil, **3** Nucleus of Research in Applied Morphology and Immunology, Morphology, Faculty of Medicine, University de Brasília, Brasília, Federal District, Brazil, **4** Department of Otolaryngology, Faculty of Medicine, University of Brasilia, Brasília, Federal District, Brazil, **5** D'Or Institute for Research and Education (IDOR), Brasília, Federal District, Brazil, **6** Department of Pediatric, Faculty of Medicine, University of Brasilia, Brasilia, Distrito Federal, Brazil

⊙ These authors contributed equally to this work.
\* glararezende@gmail.com

**Data Availability Statement:** All relevant data are within the manuscript and its Supporting information files.

## Abstract

The role of phagocytes of children with cystic fibrosis (CF) associated with different phenotypes of chronic rhinosinusitis (CRS) is unclear. The aim of this study was to evaluate the phagocytic capacity of blood neutrophils and monocytes and production of superoxide anion by phagocytes in patients with CF with or without chronic rhinosinusitis and with or without nasal polyps (NP). This cross-sectional study was established in 2015–2017 in a tertiary reference center to the CF treatment, Brasilia, Brazil. Sample included 30 children volunteers with CRS related to CF (n = 16) and control subjects (n = 14). Epidemiological and clinical data were compared. Collection of 15 mL of peripheral blood and nasal endoscopy to identify the presence or absence of nasal polyps (NP) were performed. Phagocytosis of *Saccharomyces cerevisiae* by pathogen-associated molecular pattern receptors and opsonin receptors was assessed. Superoxide anion production was evaluated. The control group showed a higher phagocytic index to monocytes and neutrophils than to the CF or CF +CRS with NP groups [Kruskal-Wallis p = 0.0025] when phagocytosis were evaluated by pathogen-associated molecular pattern receptors (5 yeasts/cell). The phagocytic index of the CF+CRS without NP group was higher than in the CF+CRS with NP group (Kruskal-Wallis p = 0.0168). In the control group, the percentage of phagocytes involved in phagocytosis and superoxide anion production (74.0 ± 9.6%) were higher in all CF groups (p < 0,0001). The innate immune response, represented by phagocytic activity and superoxide anion production by monocytes and neutrophils was more impaired in patients with CF related or not related to CRS than in the control group. However, the phagocytic function of patients without NP showed less impairment.

**Funding:** The material used in this research came from the Cellular Immunology Laboratory's own resources provided by Faculty of Medicine, University of Brasilia and self-donation of authors. The funders had no role in study design, data collection and analysis, decision to publish, or preparation of the manuscript. The authors received no specific funding for this work.

**Competing interests:** The authors have declared that no competing interests exist.

## Introduction

Cystic fibrosis (CF) is characterized by an excessive inflammatory response and the failure to efficiently resolve lung infections, causing major morbidity and mortality [1]. The disease may involve chronic rhinosinusitis (CRS) and nasal polyps (NP). Cystic fibrosis (CF) related CRS causes deficiencies in the phagocytic activity of macrophages and neutrophils [2–5]. However, it remains unclear whether these changes occur because of the expression of local inflammatory factors or are primary defects depending on CF transmembrane conductance regulator (CFTR) in these leukocytes [6–10]. NP in children with CRS related to CF is considered as a complication resulting from inflammatory changes in the host [11] or may be a primary manifestation of CF by the inflammatory process before infection, indicating that expression of CFTR has a functional role in monocytes and neutrophils [6–10].

Local inflammatory changes influence mucociliary clearance and the ability of phagocytes to eliminate invading microorganisms by increasing $Na^+$ and $Cl^-$ concentrations in the cell cytosol. A reduction of $Cl^-$ in lysosomes [12–16] is the main trigger of inflammatory processes in CF, which activate the M1 type of macrophages in CF-NP [17]. CF-NP is associated with upregulation of human β-defensin 2 and Toll-like receptor 2, high tissue infiltration of neutrophils [17] and levels of oxygen species [11]. In patients with CF without NP, expression of the macrophage mannose receptor dominates the innate defense. The inflammation of respiratory tissues leads to the plasticity of macrophage and neutrophil phagocytic activity between a pro-inflammatory, immunogenic, or tissue destructive status or to an anti-inflammatory and tolerogenic phenotype [11].

The genetic basis of CF has been clearly established [17, 18], but no specific mutation is correlated with the impact on the phagocytic activity of neutrophils and monocytes [7, 11] and CF polyp phenotypes. CFTR-like defect has been detected in human monocytes, suggesting that CF macrophage and neutrophil dysfunction is a partial consequence of CFTR defects [1, 10, 19]. Moreover, inhibition of functional CFTR in wild-type macrophages was shown to lead to a phenotype similar to that in CF macrophages [7].

This study was conducted to assess the innate immunity of patients with different CF subtypes. By examining phagocytosis and superoxide anion production [20–23], we evaluated whether neutrophils and monocytes that circulate in the peripheral blood participate indirectly in inflammatory activity in the paranasal sinuses [6–8].

## Materials and methods

### Patient demographics

This study followed the ethical standards for scientific research on humans in accordance with Law 6,638/79 and was conducted according to the guidelines prescribed by the Declaration of Helsinki [24]. The study was approved by the local Research Ethics Committee (Protocol No. 085/2010). All volunteers and their parents or guardians signed written consent agreeing to the research.

The study included 36 children; their demographic data were collected through a questionnaire at the time of enrollment in a pneumological pediatric clinic of a tertiary hospital in Brazil. The questionnaire assessed the age, gender, presence of nasal symptoms, nutritional status and bacterial colonization status. The nutritional status was classified as a percentile according to the age and normal weight (≥3 and ≤97), malnourished (<3) and overweight (>99,9) [25]. All patients lived in the central area of Brazil. Inclusion criteria were a diagnosis of CF, children >8 or <18 years, and not using any drug except those administered to treat CF. CF was confirmed in children whose chloride level in the sweat test was higher than 60 mmol/L[1]. The

**Table 1. Demographics and characterization of study participants.**

| Groups | Age (years) mean ± SD | Pharynx swab (<3 months) | Previous surgery | Nutritional status | Digital clubbing | Lund Kennedy endonasal Scale |
|---|---|---|---|---|---|---|
| Control (n = 14) | 10 ± 2 | - | - | normal weight | no | 0.0 ± 0.0 |
| Cystic fibrosis (n = 5) | 9 ± 4 | *Pseudomonas* | - | normal weight | no | 0.4 ± 0.2 |
| | | *Pseudomonas* | - | malnourished | yes | |
| | | *Pseudomonas* | - | malnourished | no | |
| | | *Bulkholderia cepacea* | - | malnourished | no | |
| | | *Pseudomonas* | - | normal weight | yes | |
| CF+CRS with NP (n = 6) | 10 ± 2 | - | ESS | normal weight | yes | 6.4 ± 2.1 |
| | | *Pseudomonas* | ESS | normal weight | no | |
| | | *Pseudomonas* | - | normal weight | no | |
| | | *Pseudomonas* | - | normal weight | yes | |
| | | *Pseudomonas* | - | normal weight | no | |
| | | *Staphylococcus* | liver transplantation# | normal weight | no | |
| CF+CRS without NP (n = 5) | 8 ± 4 | *Staphylococcus* | - | malnourished | yes | 0.4 ± 2.2 |
| | | *Pseudomonas* | - | normal weight | no | |
| | | *Staphylococcus* | - | malnourished | no | |
| | | *Pseudomonas* | - | malnourished | no | |
| | | *Pseudomonas* | - | overweight | no | |

CF = Cystic fibrosis; CRS = chronic rhinosinusitis; NP = nasal polyps; ESS: endoscopic sinus surgery; n = 30

# = 8th year after liver transplantation because of congenital biliary atresia, without immunosuppressive drugs at the moment

test was performed by placing a pilocarpine solution on the forearm or thigh and stimulating the area with a mild electric current to produce sweating. Subjects with acute respiratory infection, diabetes, an autoimmune disease, severe allergies, renal or gastrointestinal diseases, cancer, anemia or any condition that was considered by the investigators to alter immune system function, including the use of medications that could change the level of inflammatory mediators, were excluded from the study. Children with CF were included in the following groups: CF plus CRS and presenting with NP (CF+CRS with NP, n = 6), CF plus CRS but without NP (CF+CRS without NP, n = 5), and CF without CRS (CF, n = 5). A group of 14 healthy children without CF and without CRS were included as the control group (Table 1).

Healthy children from the control group were recruited from the pediatric clinic and did not present with ear, nose, or throat problems and did not have a family history of CF.

The inclusion criterion for CRS followed the classification based on the European Position Paper on Rhinosinusitis and Nasal Polyps 2020 [2]. CRS was defined as two or more symptoms for over 12 weeks, one of which was nasal blockage or nasal discharge, and ± facial pain/ pressure, ± reduction, or loss of smell or ± cough (children). In addition, endoscopic signs of NP and/or mucopurulent discharge from the middle meatus were necessary to define CRS. All subjects with CF underwent rigid nasal endoscopy examination (zero-degree endoscope, Fiegert Endotech, Fort Lauderdale, FL, USA) and were classified following the methods and criteria proposed by Lund and Kennedy [26] (patients scored using a scale of 0–2 points based on the presence of polyps, edema, discharge, scarring, and crusting).

The participants underwent general clinical and ear, nose, and throat examination, and were subjected to nasal endoscopy. Peripheral blood was collected into 3 sterilized, 5-mL vacuum tubes to measure the phagocytic activities of monocytes and neutrophils as

primary outcomes. The secondary outcome was the production of superoxide anion in each group.

## Phagocytosis test

Phagocytosis of *Saccharomyces cerevisiae* was evaluated as previously reported by Muniz-Junqueira et al. with some modifications [21]. Briefly, samples of 40 μL/area of whole peripheral blood were placed on clean glass slides containing 8 marked areas of 7-mm diameter each in duplicate preparations. The slides were incubated in a wet chamber for 45 min at 37˚C and then rinsed with 0.15 M phosphate-buffered saline pH 7.2 at 37˚C to remove non-adherent cells. After washing, the neutrophils and monocytes remained adhered to the slide in approximately the same proportion as that observed in whole blood (12,534 ± 5050 cells/marked area; 5.63 ± 0.85% monocytes and 93.5 ± 1.08% neutrophils). Adhered cells were incubated with a suspension of $6.5 \times 10^4$ (1/5 ratio phagocyte/*S. cerevisiae*) or $2.5 \times 10^5$ *S. cerevisiae* yeast (1/20 ratio phagocyte/*S. cerevisiae*) in 20 μL of Hanks-Tris solution (Sigma, St Louis, MO, USA) containing 10% heat-inactivated fetal calf serum (Gibco, Grand Island, NY, USA), pH 7.2, in a wet chamber at 37˚C, for 30 min. To evaluate the influence of complement molecules on phagocytosis, *S. cerevisiae* were sensitized by incubation, at 37˚C in 10% fresh serum obtained from the donor samples in Hanks-Tris solution for 30 min. The slides were rinsed with 0.15 M phosphate-buffered saline at 37˚C to eliminate non-phagocytosed *S. cerevisiae*. After washing with Hanks-Tris containing 30% fetal calf serum, the slides were fixed with absolute methanol and stained with Giemsa solution (10%). The phagocytic index (PhI) of 200 monocytes and 200 neutrophils in individual preparations was assessed by optical microscopy (1000x; CH30, Olympus, Tokyo, Japan). Microscopic fields distributed throughout the slide were randomly selected, and all monocytes and all neutrophils in each field were blinded and examined by the same observer. The PhI was calculated as the mean number of phagocytosed *S. cerevisiae* per phagocytosing monocyte or neutrophil, multiplied by the percentage of phagocytes engaged in phagocytosis [21, 22].

The baking yeast (*S. cerevisiae*) suspension was prepared as described previously [20, 21, 23]. *Saccharomyces cerevisiae* yeasts are phagocytosed via receptors. Phagocytosis may occur via pattern recognition receptors when receptors recognize directly conserved pathogen-associated molecular patterns (PAMPs) on the surface of the particle to be phagocytosed. In addition, phagocytosis may be facilitated by opsonins when internalization of the particle occurs through receptors to components of complement or via receptors to FcIgG [23]. *Saccharomyces cerevisiae* were used with or without previous incubation with fresh serum from the donor. In the first case, yeasts were opsonized by complement molecules and antibodies present in the fresh serum, which adhered on the yeast surface and phagocytosis occurred by complement receptors or by receptors to the Fc portion of immunoglobulin G (IgGFc receptors). When the yeasts were incubated with fetal calf serum, phagocytosis occurred via the pattern-recognition receptors of monocytes and neutrophils [23].

## Production of superoxide anion

The nitro blue tetrazolium (NBT) salt reduction method [23] was used to evaluate the production of superoxide anion (O⁻). This reactive oxygen species reduces the NBT compound to an insoluble form, formazan, which is visualized by optic microscopy as a blue color in the cytoplasm of phagocytes.

Phagocytes adhered on the slide as described above were incubated with 0.05% NBT in Hanks-Tris (Sigma) solution for 20 min in a wet chamber at 37˚C (baseline O⁻ production). Stimulated superoxide anion production by phagocytes was evaluated after a suspension of *S.*

*cerevisiae* was added at a ratio of 1 phagocyte/5 yeast per well and incubated for 20 min in a wet chamber at 37°C. The slides were washed, fixed with methanol, and stained with a solution of 1.4% safranin and 28.6% glycerol in distilled water. The results are expressed as the percentage of phagocytes (monocytes plus neutrophils) that reduced the NBT salt, and the percentage of reduction of NBT was quantified by optical microscopy individually for each patient. The total percent of phagocytes that reduced the NBT, percent of phagocytes phagocytosing *S. cerevisiae* that reduced or did not reduce NBT, and percent of phagocytes that were not in phagocytosis and reduced or did not reduce NBT were determined.

## Statistical analysis

The results were evaluated by Bartlett's test for equal variances and the Kolmogorov–Smirnov test for normally distributed data before comparative analysis. The data were evaluated by analysis of variance (ANOVA), followed by the Student–Newman-Keuls method or Kruskal-Wallis test and Dunn's method to compare multiple unrelated samples of normally or non-normally distributed data, respectively. The Student's t test was performed to compare two normal unrelated samples. For non-normal distributions, the Mann-Whitney test was used to compare two unrelated groups. GraphPad Prism 8.0 software (GraphPad, Inc., La Jolla, CA, USA) was used for statistical tests and graphical presentation of the data. Differences with a two-tailed value of $p < 0.05$ and 95% confidence intervals (CI) were considered as statistically significant.

## Results

The demographic characteristics of the patients are shown in Table 1. Six patients who presented with acute respiratory infection at the examination and/or with non-specific CRS were excluded from the study group. Therefore, the investigation included 30 subjects. No correlation was observed between nutritional status and phagocytosis tests (Pearson correlation; $p > 0.05$).

### Phagocytic capacity of neutrophils and monocytes

**Phagocytosis by PAMPs.** The percentage of neutrophils involved in phagocytosis was higher in the control group (14.8%) [95% confidence interval (CI): 10.44–27.78] than in the CF (1.5%, CI: -2.22–9.02) and in CF+CRS with NP (1.5%, CI: -1.77–9.83) groups [Kruskal-Wallis (KW); p = 0.0025 (5 yeast/cell)]. In a more concentrated proportion of yeasts (20 yeasts/neutrophil), the percentage of neutrophils was more involved in phagocytosis in the control group (26%, CI: 12.47–33.45) than in CF+CRS with NP (1.5%, CI: 0.07–1.524) (KW, p = 0.0005). Furthermore, the percentage of neutrophils engaged in phagocytosis in the CF+CRS without NP group (4.5%, CI: 3.33–5.36) was higher than in the CF+CRS with NP group (0.5%, CI: 0.07–1.524) (KW = 0.028) (Fig 1A1 and S1 Table). Moreover, the number of yeasts phagocytosed by neutrophils was higher in the control group (1.4, CI: 1.24–1.55) than in the CF group (1, CI: 0.96–1.07). In addition, the number of yeasts phagocytosed by neutrophils in CF (1, CI: 0.96–1.07) and CF+CRS with NP (1, CI: 0.42–1.70) were lower than in CF+CRS without NP (1.9, CI: 0.86–1.20) (KW = 0.048) (Fig 1A2). The PhI was also higher in the control group (18.8, CI:14.37–10.16) than in the CF (1.5, CI: -2.76–10.16) and CF+CRS with NP groups (2, CI: -1.3–10.22) (KW = 0.0014) in the evaluation of 5 yeasts/cell. When this was tested using 20 yeasts/cell, the PhI of the control group (20.8, CI: 13.66–37.67) was higher than the CF+CRS with NP group (1.5, CI: 0.25–2.51) (KW = 0.0128). In addition, the PhI was higher in the CF+CRS without NP group (5, CI: 0.87–8.32) than in the CF (1.5, CI: -1.71–5.21) and CF+CRS with NP groups (2, CI: 0.25–2.51) (KW = 0.0168) (Fig 1A3 and S1 Table).

We found no difference among groups when phagocytosis by monocytes was tested through the PAMPs (KW, $p > 0.05$) (Fig 2 and S1 Table).

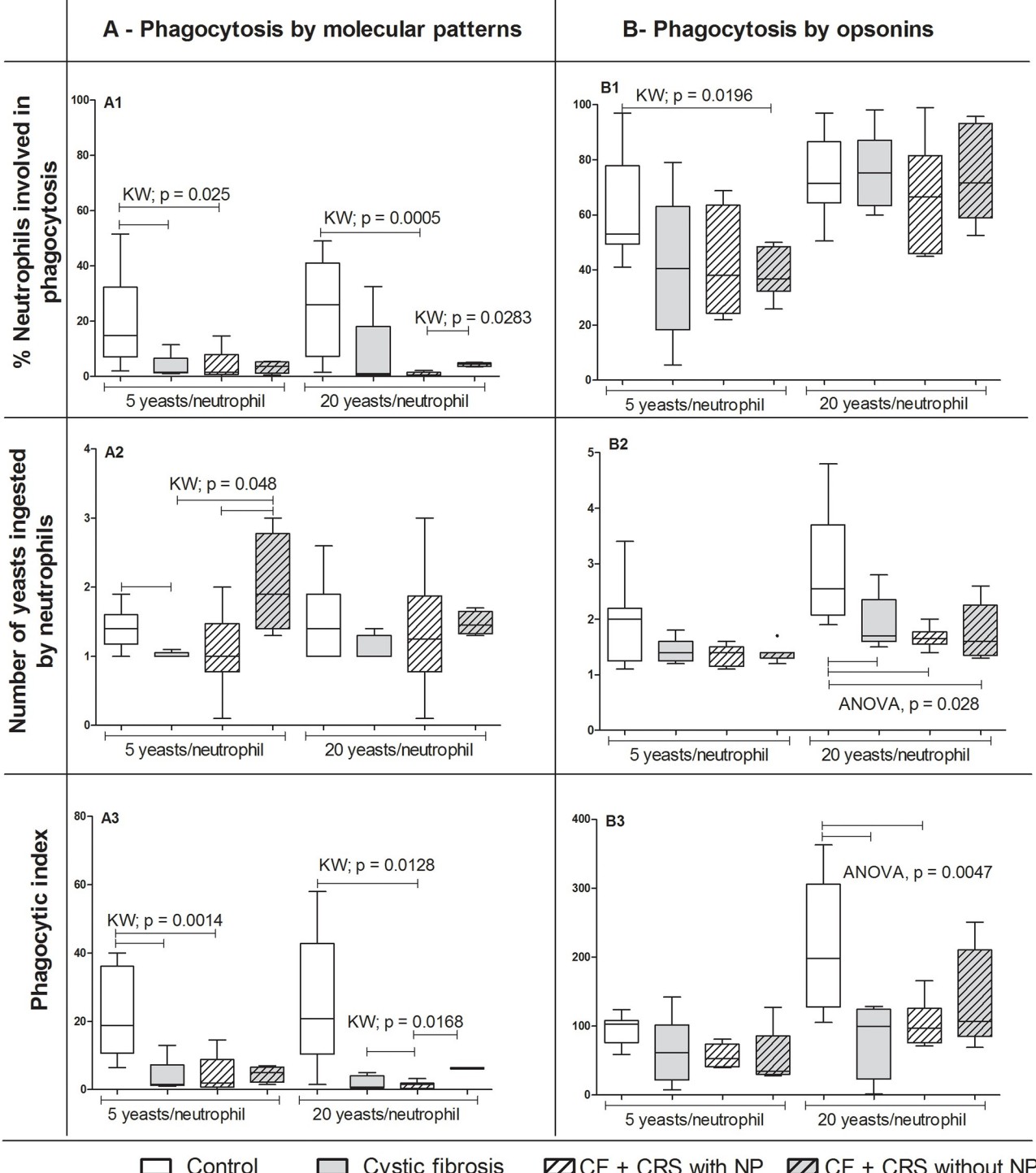

**Fig 1. Phagocytosis by neutrophils.** Phagocytosis of *Saccharomyces cerevisiae* cells by pathogen-associated molecular pattern A) and opsonin receptors (B) by blood peripheral neutrophils in children. Groups: cystic fibrosis (CF) n = 6; cystic fibrosis plus chronic rhinosinusitis and presenting nasal polyps (CF+CRS with NP) n = 6; cystic fibrosis plus chronic rhinosinusitis but without nasal polyps (CF+CRS without NP) n = 5; control group (n = 14). In the top, % of neutrophils involved in phagocytosis, in the middle, number of yeasts ingested by neutrophils and in the bottom phagocytic index. Significant results are shown by bar between groups. Values are presented as the median, quartiles, maximum, and minimum values. KW = Kruskal-Wallis.

**Phagocytosis facilitated by opsonins.** For neutrophils, we observed a higher percentage of cells involved in phagocytosis (5 yeast/cell) in the control group (53%, CI: 51.74–73.55) than in the CF+CRS without NP group (36.8%, CI: 29.11–48.16) (KW, p = 0.0196) (Fig 1B1 and S2 Table). A reduced number of yeasts ingested by neutrophils (20 yeasts/cell) was also observed in all groups [CF = 1.7 (1.28–2.56); CF+CRS with NP = 1.7 (1.46–1.83); CF+CRS without NP = 1.6 (1.11–2.4)] compared to the control group = 2.6 (2.35–3.44) (ANOVA, p = 0.028) (Fig 1B2 and S2 Table). Moreover, the control group (198; CI: 167.4–272.4) showed a higher PhI than the CF (99.2; CI: -7.59–171.8) and CF+CRS with NP groups (97.1; CI: 67.32–139.4) (ANOVA, p = 0.0047) when tested using 20 yeasts/cell (Fig 1B3 and S2 Table).

For monocytes, we observed a larger number of yeasts ingested by monocytes (20 yeast/cell) in the control group (2.6; CI: 2.33–3.49) than in the CF+CRS with NP group (2.1; CI: 1.51–2.32) (KW = 0.0375 and t test) (Fig 2B2 and S2 Table). Evaluation of the PhI of monocytes facilitated by opsonins showed that the control group (102.5; CI: 82.83–107.4) had a higher PhI (5 yeasts/monocyte) than in all other groups [CF = 24.5 (15.9–81.18); CF+CRS with NP = 55.8 (9.27–78.56); CF+CRS without NP = 19.6 (13.33–44.86)] (ANOVA, p < 0.001) (Fig 2B3 and S2 Table).

## Evaluation of superoxide anion production by phagocytes

**Percentage of phagocytes producing superoxide anions at the basal production.** The KW test followed by Dunn's method showed that superoxide anion production by phagocytes obtained from participants with CRS with NP (43.5%; CI: 12.16–63.04) or CF (20.0%; CI: 6.310–51.29) had a lower median superoxide production than those produced by healthy participants (83.0%; CI: 66.69–89.24) (p = 0.0012). In contrast, there was no difference between the control and CRS without NP group (38.9%; CI: 19.65–68.61) (p > 0.05) (Fig 3A1).

**Percentage of phagocytes involved in phagocytosis without production of superoxide anion.** The median number of phagocytes involved in phagocytosis without $O^-$ production in all groups [CF = 31.5% (CI: 1-.61–67.43), CF+CRS with NP = 21.5% (CI: 7.91–34.05) or CF+CRS without NP = 27.5% (CI: 3.37–56.96)] was higher than that in the control group = 0.0% (p = 0.0001, CI: 0.003–1.817) (Fig 3B1).

**Percentage of phagocytes involved in phagocytosis with superoxide anion production.** The percentage of phagocytes involved in phagocytosis that produced superoxide anion was higher in the control group (74.0 ± 9.6%; CI: 68.14–79.78) than those observed in the CF (5.9 ± 4.1%; CI: 0.87–11.01), CF+CRS with NP (2.8 ± 3.7%; CI: 1.78–7.388), and CF+CRS without NP (9.2 ± 7.6%; CI: 1.198–17.20) groups (ANOVA; p < 0.02001) (Fig 3B2).

**Percentage of phagocytes without phagocytosis or superoxide anion production.** The percentage of phagocytes that did not exhibit phagocytosis nor superoxide anion production in the control group (1.5%, CI: 1.01–4.34) was lower than that in the other groups [CF = 35% (CI: 11.92–60.12); CF+CRS with NP = 41% (CI: 16.09–70.55); CF+CRS without NP = 21.75% (CI: 5.013–54.15)] (KW; p < 0.0001) (Fig 3C1).

**Percentage of phagocytes without phagocytosis but produced superoxide radical.** The median percentage of phagocytes that were not involved in phagocytosis but produced superoxide anions did not differ among groups [control = 23.25% (CI: 18.14–25.94); CF = 10.5% (CI: 2.97–34.70; CF+CRS with NP = 34.25% (CI: 12.08–54.29); CF+CRS without NP = 16.8% (CI: 1.91–60.25) (Fig 3C2).

## Discussion

Our data showed that CF significantly impacts the phagocytosis rate and superoxide anion production by peripheral blood-derived neutrophils and monocytes, as determined by phagocytosis assay with *S. cerevisiae* [21–23] and superoxide anion production tested by NBT

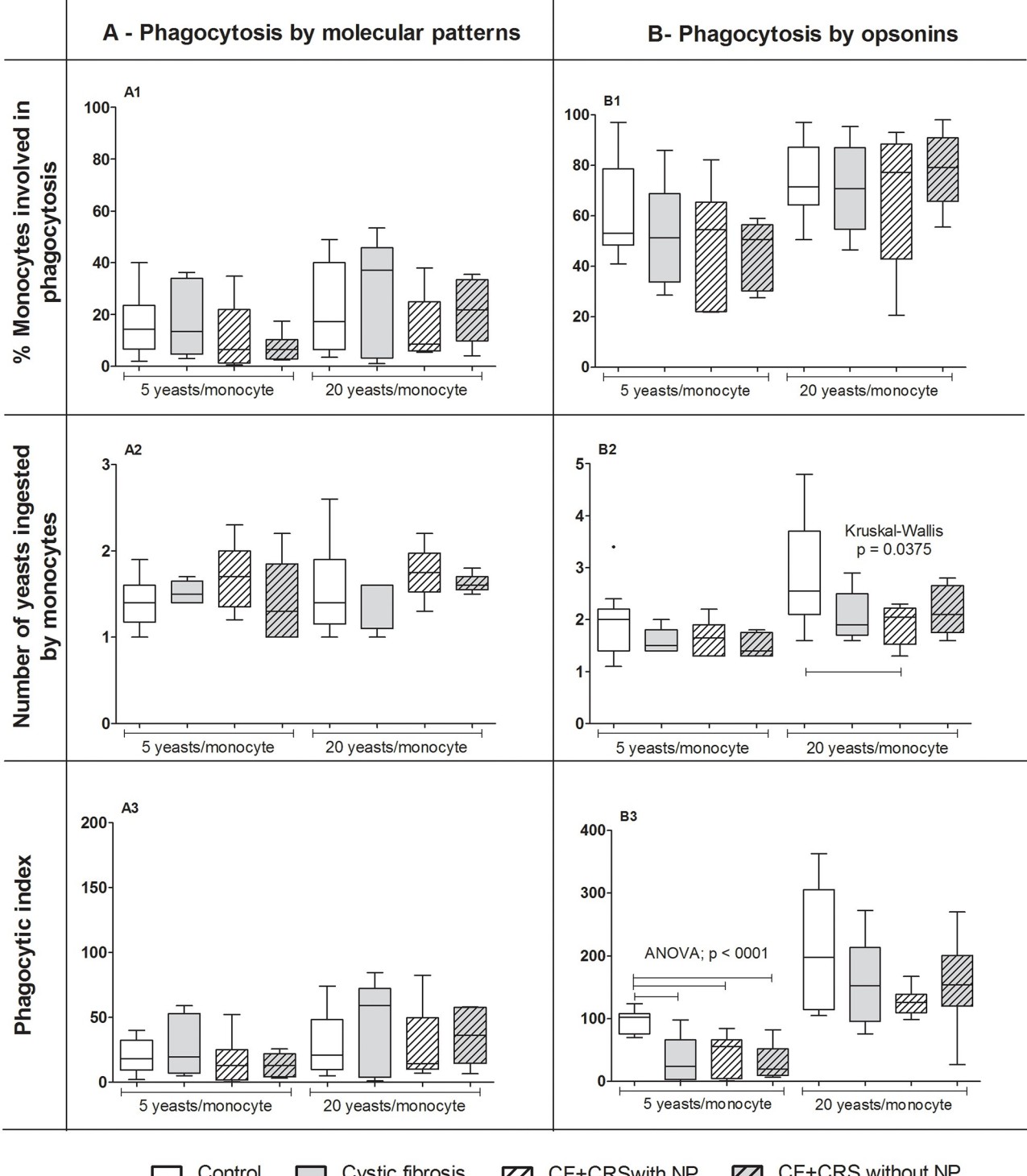

**Fig 2. Phagocytosis by monocytes.** Phagocytosis of *Saccharomyces cerevisiae* cells by pathogen-associated molecular pattern (A) and opsonin receptors (B) by blood peripheral monocytes in children. Groups: Cystic fibrosis (CF) n = 6; cystic fibrosis plus chronic rhinosinusitis and presenting nasal polyps (CF+CRS with NP) n = 6; cystic fibrosis plus chronic rhinosinusitis but without nasal polyps (CF+CRS without NP) n = 5; control group (n = 14). In the top, % of neutrophils involved in phagocytosis, in the middle, number of yeasts ingested by neutrophils and in the bottom, phagocytic index. Significant results are shown by bar between groups. Values are presented as the median, quartiles, maximum, and minimum values.

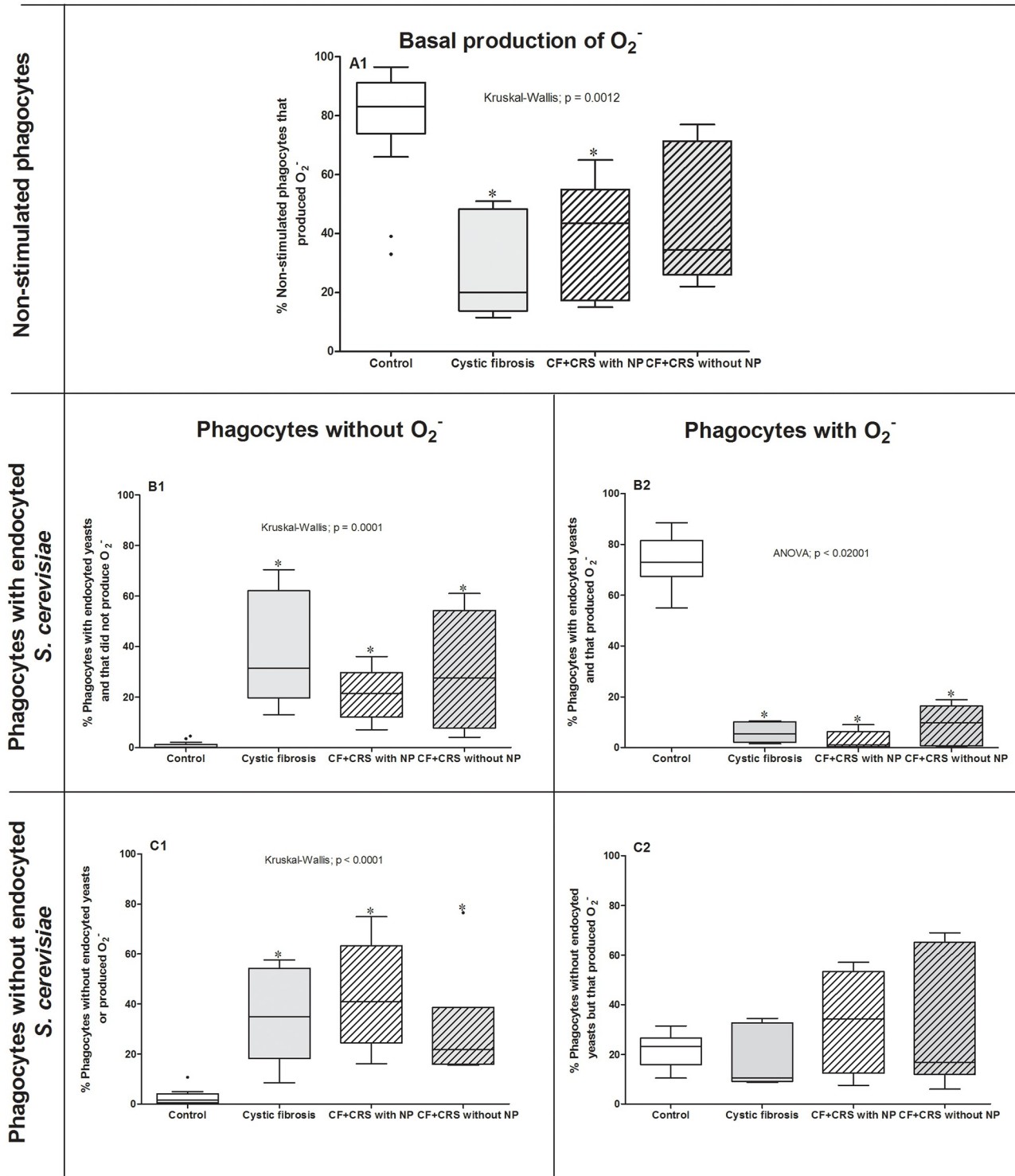

**Fig 3. Superoxide anion production.** Percentage of reduction of nitro blue tetrazolium by phagocytes obtained from peripheral blood of the control group (n = 14), cystic fibrosis group (CF; n = 5), cystic fibrosis + chronic rhinosinusitis with polyps group (CF+CRS with NP; n = 5), and cystic fibrosis + rhinosinusitis without polyps group (CF+CRS without NP; n = 6). Phagocytes were incubated with and without *Saccharomyces cerevisiae* yeasts to either stimulate or not stimulate superoxide anion production. Significant results are shown by * between groups. Values are presented as the median, quartiles, maximum, and minimum values.

reduction. This is the first study to compare this type of activity in neutrophils and monocytes in peripheral blood from patients with different CF CRS phenotypes. Because of the difficulty of investigating cell populations in the mucosa of the paranasal sinuses, neutrophils and monocytes originating in the bone marrow may reveal the behavior of leukocytes in the upper airways. Moreover, these results support that deficient CFTR is expressed in myeloid-derived cells [7].

We showed that neutrophils from patients with CF with or without CRS had lower PhIs both by lower engagement of neutrophils in phagocytosis and smaller number of phagocytosed yeasts. Interestingly, patients with CRS without NP showed an increased number of yeasts phagocytosed by the cells. According to Downey et al [27], comparison of gene expression in blood neutrophils from patients with CF and healthy controls showed that 62 genes were upregulated. However, none of these genes coded for adhesion molecules (ICAM-1 and 2), which may explain the lower capacity for yeast endocytosis by phagocytes when PAMPs and opsonins receptors were evaluated, as observed in our study. In addition, Bonfield et al[7] showed that mice myeloid cells lacking CFTR failed to control *Pseudomonas aeruginosa* infection in animal models compared to that in normal CFTR groups. In fact, *P. aeruginosa* depends mainly on phagocytosis by neutrophils for its defense, which was impaired in all of our CF patients. These findings highlight that the innate inflammatory process may begin before infection, potentially as a result of the early clinical presentation of NP observed in children, even without *P. aeruginosa* colonization. These impaired neutrophil functions indicate that a primary defect in CF leukocytes plays a major role in the clinical phenotypes of CRS.

Other defects in the phagocytic activity of these cells in individuals with CF-related CRS may be related to the presence of intramucosal microcolonies in the nasal cavity, which are mainly formed by *Staphylococcus aureus*. Few inflammatory cells exist around these microcolonies [5]. This biofilm behaves as a less immunogenic bacterial surface [5], which may be associated with the lower phagocytosis capacity of phagocytes by opsonin receptors in patients with CF with CRS than in the control group, as per observations in peripheral blood samples.

CRsNP, CRSwNP and CF-NP are different disease entities with distinct cytokine, mediator, and cellular profiles [18, 28]. Among the different types of CRS, our data showed that neutrophils from patients with CF+CRS without NP endocytosed more yeasts than patients with CF and CF+CRS with NP. CRS related to CF is associated with upregulation of β-defensin 2 and Toll-like receptor 2. Furthermore, expression of the mannose receptor dominates the innate defense in non-CF NP [18]. The significantly different outcomes between the CRS groups in CF raise new questions about the pathophysiology of NP. Neutrophils and monocytes are plastic cells, and their physiological functions partially depend on their origin [10]. Our sample of 38% patients with CF with NP differed from the results of Schmitt et al [29] but agreed with those of Weber et al [30] observed in a Brazilian population. In addition to genetic factors, diversity in geographic, socioeconomic, and nutritional status may influence the innate immunity of subjects with CF [28]. Other studies of our population are necessary to support the plasticity of these leukocytes in the peripheral blood without demographic and socioeconomic bias.

In the absence of effective phagocytosis, some phagocytes showed enhanced superoxide anion production, suggesting that in addition to phagocytosis deficiency, which prevents pathogen clearance, inadequate superoxide anion production without effective phagocytosis may enhance tissue lesions caused by inadequate enhancement in radical oxygen production [23]. Our data showed that these unbalanced responses resulted in higher production of superoxide anions without phagocytosis in all CF studied (Fig 3C1). This pro-inflammatory status was also observed as the higher percentage of *S. cerevisiae* cell ingestion. Radical oxygen species have various antibacterial, antifungal, and antiviral effects, although some of these substances

are tonically secreted [23] and are upregulated even in the absence of pathogens. In the control group, the percentage of phagocytes that participated in phagocytosis with the production of superoxide anions was higher, demonstrating a more balanced function of phagocytes. Our results suggest that superoxide anions generated by phagocytes play an essential role in upper airway innate immunity in CF, without significant differences between CRS phenotypes.

A limitation of our study was the small number of participants. Furthermore, the presence of some comorbidities and liposoluble vitamins and hemoglobin levels were not controlled. In addition, the severity of lung disease and nutritional status were only evaluated by classifying the digital clubbing and percentile by age, respectively. Although no difference in phagocytic function was observed in malnourished children, a more detailed analysis involving chest scans and determination of the corporal composition, such as the muscle and fat body distribution, may reveal a correlation with phagocytosis in these individuals.

In conclusion, patients with CF with and without CRS, exhibited impairments in phagocytosis by neutrophils and monocytes. Strategies for improving myeloid cell health and function are needed. Future research on innate immunity diagnoses and treatment is necessary.

## Supporting information

**S1 Table. Phagocytosis of *Saccharomyces cerevisiae* cells by pathogen-associated molecular pattern receptors in the peripheral leukocytes of children; values are expressed as median.**
(DOCX)

**S2 Table. Phagocytosis of *Saccharomyces cerevisiae* by opsonin receptors in the peripheral leukocytes of children; the values are expressed as median values.**
(DOCX)

**S1 File.**
(PDF)

**S2 File.**
(PDF)

**S3 File.**
(PDF)

## Author Contributions

**Conceptualization:** Gustavo L. Rezende, Marcio Nakanishi, Selma A. S. Kückelhaus, Maria I. Muniz-Junqueira.

**Data curation:** Gustavo L. Rezende, Maria I. Muniz-Junqueira.

**Formal analysis:** Gustavo L. Rezende, Marcio Nakanishi, André L. L. Sampaio, Selma A. S. Kückelhaus, Maria I. Muniz-Junqueira.

**Investigation:** Gustavo L. Rezende, Marcio Nakanishi, Shirley C. P. Couto, Carmen L. F. S. Martins, Lucas F. F. Albuquerque, Selma A. S. Kückelhaus, Maria I. Muniz-Junqueira.

**Methodology:** Gustavo L. Rezende, Marcio Nakanishi, Shirley C. P. Couto, Selma A. S. Kückelhaus, Maria I. Muniz-Junqueira.

**Project administration:** Gustavo L. Rezende, Selma A. S. Kückelhaus, Maria I. Muniz-Junqueira.

**Resources:** Gustavo L. Rezende, Selma A. S. Kückelhaus, Maria I. Muniz-Junqueira.

**Supervision:** Maria I. Muniz-Junqueira.

**Validation:** Gustavo L. Rezende, Maria I. Muniz-Junqueira.

**Visualization:** Gustavo L. Rezende, Maria I. Muniz-Junqueira.

**Writing – original draft:** Gustavo L. Rezende, André L. L. Sampaio, Selma A. S. Kückelhaus, Maria I. Muniz-Junqueira.

**Writing – review & editing:** Gustavo L. Rezende, Marcio Nakanishi, André L. L. Sampaio, Selma A. S. Kückelhaus, Maria I. Muniz-Junqueira.

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
