## [Decision Letter · Decision Letter 0]

24 Mar 2022

PONE-D-21-40067Alterations in innate immune responses of patients with chronic rhinosinusitis related to cystic fibrosisPLOS ONE

Dear Dr. Rezende,

Thank you for submitting your manuscript to PLOS ONE. After careful consideration, we feel that it has merit but does not fully meet PLOS ONE’s publication criteria as it currently stands. Therefore, we invite you to submit a revised version of the manuscript that addresses the points raised during the review process.

We believe addressing the reviewer's comments will significantly improve the quality and readability of the manuscript. **Please note that the detail comments of Reviewer # 1 are an attachment (Word file). Please download the comments and address them. **

We look forward to receiving your revised manuscript.

Kind regards,

Mrinmoy Sanyal, PhD

Academic Editor

PLOS ONE

Journal Requirements:

“Unfunded study”

Reviewers' comments:

Reviewer's Responses to Questions

**Comments to the Author**

1. Is the manuscript technically sound, and do the data support the conclusions?

Reviewer #1: Yes

Reviewer #2: Partly

2. Has the statistical analysis been performed appropriately and rigorously? 

Reviewer #1: Yes

Reviewer #2: Yes

3. Have the authors made all data underlying the findings in their manuscript fully available?

Reviewer #1: Yes

Reviewer #2: Yes

4. Is the manuscript presented in an intelligible fashion and written in standard English?

Reviewer #1: Yes

Reviewer #2: No

5. Review Comments to the Author

Reviewer #1: The findings in the manuscript has been well presented, providing statistical analyses as well as clear explanation of the methods implemented. Overall the manuscript is easy to follow. I have attached my comments for further perusal to help improve the manuscript.

Reviewer #2: REVIEW SUMMARY

In this study Rezende et al. aims to understand how the innate immune system’s function changes in patients of cystic fibrosis and compared it with a sub-group who have developed chronic rhinosinusitis. This manuscript was assessed for the 7 criteria for publication in PLoS One. Rezende et al. present results of original research that have not been published elsewhere. Experiments performed, statistical analyses meet high technical standard and are described in sufficient detail. The analytical rigor of the authors is demonstrated by the fact that they perform corrections for normally and non-normally distributed data. The authors also adhere to reporting and ethics standards — all volunteers and their parents/guardians signed consent forms.

However, there are deficiencies in conclusions drawn from the results, and on article presentation/reporting. The authors over-extend when drawing conclusions from their results. In the introduction section in the abstract, instead of focusing on the gap in knowledge statements about the field of cystic fibrosis is made that is unambiguous and not entirely accurate. In addition, the results section and figure panels are structured in a difficult to interpret manner.

As such, in its current form this manuscript cannot be recommended for publication. Detailed comments are below.

MAJOR COMMENTS ON CONCLUSION

1. In the introduction section (Page 7, last paragraph), the authors write “… we evaluated whether neutrophiles and monocytes that circulate in the peripheral blood participate directly in inflammatory activity in the paranasal sinuses”. This is incorrect. In vitro assessment of phagocytotic activity and superoxide anion production of neutrophiles and monocytes cannot be used to determine whether these cells participate directly in inflammatory activity in the paranasal sinuses. This is particularly true because these immune cells were isolated from peripheral blood collected in vacuum tubes, presumably through venous blood draw from the arm (Materials and Methods section, Page 9, 3rd paragraph). This conclusion is incorrect.

2. In the highlights section (Page 3, 2nd bullet point), the authors write “Chronic Rhinosinusitis related to Cystic Fibrosis group has impaired leukocytes compared to control group”. This is an over-extension of the results. The authors measure in vitro phagocytotic ability of neutrophiles and monocytes (Y-axis labels, Fig 1 and 2), not for leukocytes that include B-cells, cytotoxic T-cells. Hence this statement is incorrect and does not accurately reflect the results.

MAJOR COMMENTS ON ARTICLE PRESENTATION/REPORTING

1. The results section is structured in a difficult to interpret manner. Numbering is not used, and nesting of sub-sections is achieved through italicizing. Also, each paragraph in the “Evaluation of superoxide production by phagocytes” section has its own sub-heading. This is not required and adds unnecessary word count to the manuscript.

2. In the abstract the authors write “The role of leukocytes in the innate immune system of children with cystic fibrosis (CF) is unclear, mainly when associated with chronic rhinosinusitis”. The sentence should be constructed in an unambiguous manner that aligns with the experiments conducted in this paper. As written, the sentence insinuates that the role of leukocytes in pediatric patients of cystic fibrosis is unknown.

MINOR COMMENTS ON ARTICLE PRESENTATION/REPORTING

1. Figures are labeled in as A1, B2. The panel naming should be alphabetical. The 3 x 2 grid structure of the figure panels is enough to segment the experimental results and orient the reader.

2. References 17, 18 and 7, 11 in the introduction section (Page 6, 3rd paragraph, first sentence) are misplaced/swapped. This compound statement is supported by two sets of references, which support the opposite halves of the sentence. “The genetic basis of CF has been clearly established, …” should be supported by references 17, 18 and not 7, 11. The opposite is true for the second part of that sentence. “… but no specific mutation is correlated with the impact on the phagocytic activity of neutrophils and monocytes” should be supported by 7, 11 not 17, 18.

6. PLOS authors have the option to publish the peer review history of their article (what does this mean?). If published, this will include your full peer review and any attached files.

Reviewer #1: No

Reviewer #2: No

---

## [Author Response · Author response to Decision Letter 0]

3 Apr 2022

Re: Manuscript Ref.: PONE-D-21-40067. “Alterations in innate immune responses of patients with chronic rhinosinusitis related to cystic fibrosis” 

Comments from authors

We are very grateful to the reviewers for their suggestions, which greatly improved the quality of our manuscript.

Reviewer 1 

“The findings in the manuscript has been well presented, providing statistical analyses as well as clear explanation of the methods implemented. Overall the manuscript is easy to follow. I have attached my comments for further perusal to help improve the manuscript.”

“The article by Gustavo Lara Rezende and colleagues, “Alterations in innate immune responses of patients with chronic rhinosinusitis related to cystic fibrosis”, offers insights on the role of monocytes and neutrophils in CF +/- CRS+/-NP and importance for the need of more diagnostic and treatments options for this illness. Overall, the study has been well-executed, and the manuscript well written providing explanations and background information to make the manuscript an enjoyable and easy to understand read.

I have minor suggestions to improve the manuscript”.

Query 1: 

Materials and Methods 

Formatting: 

Pg.9 – ‘In addition, endoscopic signs… meatus or were necessary to define CRS’. The word “OR” should be deleted.

Answer 1: 

Thank you, “OR” was deleted. 

Query 2: 

Pg. 10 – Second paragraph… ‘Saccharomyces cerevisiae yeast are phagocytosed via receptors. Phagocytosis…. Align the paragraph accordingly.

Answer 2: 

The paragraph was aligned. 

Query 3. 

Table 1: As transplant patients are given immunosuppressive (IS) drugs, it would be nice to mention the # of years post-transplant for the specific subject under the CF+CRS with NP group and clearly specify whether this subject was still on any IS drugs or not to infer that any alterations in this subject’s immune system was not contributed from post-transplant treatment. 

Answer 3: Thank you for this important observation. It was added: “8th year after liver transplantation because of congenital biliary atresia, without immunosuppressive drugs at the moment.” Please, see in Table 1 

Query 4: 

Figures 1 and 2: 

It would be better to rename A, B and C as top, middle and bottom panel respectively to avoid confusion with the figure names A1, A2, etc. and as “A, B and C” are not stated on the figures.

Answer 4:

 In the figures 1 and 2, it was included the letter “A” for phagocytosis by molecular patters and “B” for phagocytosis by opsonins. In the legend, the figures were identified by “top” for % of neutrophils involved in phagocytosis, “middle” for number of yeasts ingested by neutrophils and “bottom” for phagocytic index. Please, see in the figures 1 and 2.

Query 5: 

Figure 1A3- under 20 yeasts/neutrophils there is an additional undefined box plot along with a statistical line after CF+CRS without NP. Please clarify. 

Answer 5: 

Thank you. This was a mistake that was now removed.

Query 6: 

Supplementary Table 1 & 2: Kindly explain what you mean by ‘*values different from their respective control’? 

Answer 6: 

It means that the values are p<0,05. It was included in the text. Please, see in S1 e 2 Tables.

Query 7:

 Supplementary Table 2: Both CF+CRS with/without NP have the same number of yeasts ingested by monocytes 2.1 (20 years per cell). Yet a statistical difference is only observed for Ctrl vs CF+CRS with NP and not Ctrl vs CF+CRS without NP. Could you provide some insights on the same?

Answer 7: 

Please, observe in the figure 2B2 that although the median is the same for both patients CP+CRS with or without NP, the box-plot for patients with NP shows the interquartile range and the minimum value below that of patients without NP. In addition, statistical test showed that the number of yeasts ingested by monocytes of control group was higher than that of CP+CRS with NP. However, this difference was not significant for the patients without NP.

Reviewer 2

“In this study Rezende et al. aims to understand how the innate immune system’s function changes in patients of cystic fibrosis and compared it with a sub-group who have developed chronic rhinosinusitis. This manuscript was assessed for the 7 criteria for publication in PLoS One. Rezende et al. present results of original research that have not been published elsewhere. Experiments performed, statistical analyses meet high technical standard and are described in sufficient detail. The analytical rigor of the authors is demonstrated by the fact that they perform corrections for normally and non-normally distributed data. The authors also adhere to reporting and ethics standards — all volunteers and their parents/guardians signed consent forms.

However, there are deficiencies in conclusions drawn from the results, and on article presentation/reporting. The authors over-extend when drawing conclusions from their results. In the introduction section in the abstract, instead of focusing on the gap in knowledge statements about the field of cystic fibrosis is made that is unambiguous and not entirely accurate. In addition, the results section and figure panels are structured in a difficult to interpret manner.

As such, in its current form this manuscript cannot be recommended for publication. Detailed comments are below.

MAJOR COMMENTS ON CONCLUSION

Query 1 

In the introduction section (Page 7, last paragraph), the authors write “… we evaluated whether neutrophiles and monocytes that circulate in the peripheral blood participate directly in inflammatory activity in the paranasal sinuses”. This is incorrect. In vitro assessment of phagocytotic activity and superoxide anion production of neutrophiles and monocytes cannot be used to determine whether these cells participate directly in inflammatory activity in the paranasal sinuses. This is particularly true because these immune cells were isolated from peripheral blood collected in vacuum tubes, presumably through venous blood draw from the arm (Materials and Methods section, Page 9, 3rd paragraph). This conclusion is incorrect.

Answer 1: 

We agree with the reviewer. It was changed the word “directly” to “indirectly”. Please, see in the introduction, last paragraph

Query 2: 

 In the highlights section (Page 3, 2nd bullet point), the authors write “Chronic Rhinosinusitis related to Cystic Fibrosis group has impaired leukocytes compared to control group”. This is an over-extension of the results. The authors measure in vitro phagocytotic ability of neutrophiles and monocytes (Y-axis labels, Fig 1 and 2), not for leukocytes that include B-cells, cytotoxic T-cells. Hence this statement is incorrect and does not accurately reflect the results.

Answer 2: We agree with the reviewer. For better comprehension, it was changed “leukocytes” to “neutrophils and monocytes”. Please, see in highlights.

MAJOR COMMENTS ON ARTICLE PRESENTATION/REPORTING

Query 3: 

1. The results section is structured in a difficult to interpret manner. Numbering is not used, and nesting of sub-sections is achieved through italicizing. Also, each paragraph in the “Evaluation of superoxide production by phagocytes” section has its own sub-heading. This is not required and adds unnecessary word count to the manuscript.

Answer 3: The results section was sub-sectioned again according to the PLOS ONE style templates. We would prefer that remain the sub-heading of subsections to make clearer the topic. These sub-sections were formatted in Level 2 headings, bold type 16pt and Level 3 headings, bold type, 14pt font as style templates instructions. 

Query 4:

2. In the abstract the authors write “The role of leukocytes in the innate immune system of children with cystic fibrosis (CF) is unclear, mainly when associated with chronic rhinosinusitis”. The sentence should be constructed in an unambiguous manner that aligns with the experiments conducted in this paper. As written, the sentence insinuates that the role of leukocytes in pediatric patients of cystic fibrosis is unknown.

Answer 4: 

Thank you for the observation. The sentence was changed to make clearer: “The role of phagocytes of children with cystic fibrosis (CF) associated with different phenotypes of chronic rhinosinusitis (CRS) is unclear.” Please see in abstract:

MINOR COMMENTS ON ARTICLE PRESENTATION/REPORTING

Query 5: 

1. Figures are labeled in as A1, B2. The panel naming should be alphabetical. The 3 x 2 grid structure of the figure panels is enough to segment the experimental results and orient the reader.

Answer 5: The identification in the legends of the figures 1 and 2 were changed “top” for % of neutrophils involved in phagocytosis, “middle” for number of yeasts ingested by neutrophils and “bottom” for phagocytic index. However, we would prefer that the labels A1, A2, etc. remain to facilitate the indication of the figures in the text in results

Query 6

2. References 17, 18 and 7, 11 in the introduction section (Page 6, 3rd paragraph, first sentence) are misplaced/swapped. This compound statement is supported by two sets of references, which support the opposite halves of the sentence. “The genetic basis of CF has been clearly established, …” should be supported by references 17, 18 and not 7, 11. The opposite is true for the second part of that sentence. “… but no specific mutation is correlated with the impact on the phagocytic activity of neutrophils and monocytes” should be supported by 7, 11 not 17, 18.

Answer 6: 

Thank you. The order of references was changed in the manuscript. Please, see in the Introduction, 3rd paragraph.

---

## [Decision Letter · Decision Letter 1]

20 Apr 2022

Alterations in innate immune responses of patients with chronic rhinosinusitis related to cystic fibrosis

PONE-D-21-40067R1

Dear Dr. Rezende,

We’re pleased to inform you that your manuscript has been judged scientifically suitable for publication and will be formally accepted for publication once it meets all outstanding technical requirements.

Kind regards,

Mrinmoy Sanyal, PhD

Academic Editor

PLOS ONE

Reviewers' comments:

Reviewer's Responses to Questions

**Comments to the Author**

1. If the authors have adequately addressed your comments raised in a previous round of review and you feel that this manuscript is now acceptable for publication, you may indicate that here to bypass the “Comments to the Author” section, enter your conflict of interest statement in the “Confidential to Editor” section, and submit your "Accept" recommendation.

Reviewer #1: All comments have been addressed

Reviewer #2: All comments have been addressed

2. Is the manuscript technically sound, and do the data support the conclusions?

Reviewer #1: Yes

Reviewer #2: Yes

3. Has the statistical analysis been performed appropriately and rigorously? 

Reviewer #1: Yes

Reviewer #2: Yes

4. Have the authors made all data underlying the findings in their manuscript fully available?

Reviewer #1: Yes

Reviewer #2: Yes

5. Is the manuscript presented in an intelligible fashion and written in standard English?

Reviewer #1: Yes

Reviewer #2: Yes

6. Review Comments to the Author

Reviewer #1: The authors has addressed all the comments of the reviewer and revised the manuscript as required.

Reviewer #2: The authors have addressed the major concerns that were raised. Regarding the points on article presentation/reporting, it would be preferable if the editorial team took that decision since they are more experienced in such decisions.

7. PLOS authors have the option to publish the peer review history of their article (what does this mean?). If published, this will include your full peer review and any attached files.

Reviewer #1: No

Reviewer #2: No

---

## [Editor Report · Acceptance letter]

28 Apr 2022

PONE-D-21-40067R1 

Alterations in innate immune responses of patients with chronic rhinosinusitis related to cystic fibrosis 

Dear Dr. Rezende:

I'm pleased to inform you that your manuscript has been deemed suitable for publication in PLOS ONE. Congratulations! Your manuscript is now with our production department. 

Kind regards, 

on behalf of

Dr. Mrinmoy Sanyal 

Academic Editor

PLOS ONE